# Dehydration and Malnutrition—Similar Yet Different: Data from a Prospective Observational Study in Older Hospitalized Patients

**DOI:** 10.3390/nu17061004

**Published:** 2025-03-12

**Authors:** Nina Rosa Neuendorff, Rainer Wirth, Kiril Stoev, Maria Schnepper, Isabel Levermann, Baigang Wang, Chantal Giehl, Ulrike Sonja Trampisch, Lukas Funk, Maryam Pourhassan

**Affiliations:** Department of Geriatrics, Marien Hospital Herne, University Hospital, Ruhr University Bochum, 44625 Herne, Germany; ninarosa.neuendorff@elisabethgruppe.de (N.R.N.); rainer.wirth@ruhr-uni-bochum.de (R.W.); kiril.stoev@elisabethgruppe.de (K.S.); maria.schnepper@elisabethgruppe.de (M.S.); isabel.levermann@elisabethgruppe.de (I.L.); baigang.wang@elisabethgruppe.de (B.W.); chantal.giehl@elisabethgruppe.de (C.G.); ulrike.trampisch@elisabethgruppe.de (U.S.T.); lukas.funk@elisabethgruppe.de (L.F.)

**Keywords:** dehydration, impaired kidney function, malnutrition, obesity, older adults

## Abstract

Background/Objectives: Dehydration and malnutrition are common conditions in older adults. Although both are regulated by different pathways, they seem to share common risk factors, such as dysphagia and dementia. Only scarce data on their co-occurrence are published. An exploratory analysis of a multicenter prospective trial on the determinants of malnutrition to evaluate the potential association between malnutrition and dehydration in older hospitalized patients was performed. Methods: Patients underwent a comprehensive geriatric assessment, their nutritional status was evaluated using the Global Leadership Initiative on Malnutrition (GLIM) criteria, and routine laboratory tests were performed, including calculated serum osmolality. Results: A total of 454 patients were included in the analysis. Of those, 45% were classified as malnourished based on MNA-SF, and 42% according to GLIM criteria. Dehydration as determined by calculated serum osmolality was present in 32%. Multivariate binomial regression analysis revealed elevated serum creatinine (*p* < 0.001) and higher body mass index (BMI) (*p* = 0.020) as predictive factors for dehydration. Overlap between dehydration and malnutrition was present in 13% of patients; malnourished patients had no higher risk for dehydration and vice versa (*p* = 0.903). Conclusions: Malnutrition and dehydration are common in hospitalized older adults but do not frequently occur together. We identified that BMI and creatinine levels are significant predictors of dehydration risk among this population. Consequently, the implementation of separate screening assessments for malnutrition and dehydration is recommended to better identify and address these conditions individually.

## 1. Introduction

The sensation of thirst and hunger changes throughout an individual’s lifetime, which increases the risk of dehydration and malnutrition among older adults [1]. Anorexia of aging [2] and loss of thirst [3] are well-described phenomena of advanced age and form the basis for developing dehydration and malnutrition if further risk factors occur. These conditions increase not only their susceptibility to stressors but also significantly elevate the risk of adverse health-related outcomes [1]. Although malnutrition and dehydration are common conditions in older adults, only scarce data exist which evaluate the prevalence and predicting factors for their co-occurrence. Results from a previous cross-sectional study among 114 patients aged 65 years and older, hospitalized at a geriatric hospital ward, showed that 49 (43%) were dehydrated, and a co-occurrence of dehydration and malnutrition was found in 30 of the 114 (26%) patients [4].

The term ‘anorexia of aging’ has been coined to describe the age-related loss of appetite and reduced food intake, which are well-known features of aging and intimately linked to frailty [5]. The underlying causes are multifaceted. From a pathophysiological point of view, delayed gastric emptying due to altered muscular tone and motility, reduced activity of visceral neurons, and altered distension of the stomach partially explain the longer-lasting satiety of older adults [5]. Furthermore, several hormonal perturbations related to aging, such as changes in circulating levels of cholecystokinin, glucagon-like peptide 1, gastric inhibitory polypeptide, and ghrelin, may further diminish appetite [5]. Other factors, such as changes in sight, smell, and taste, poor dentition, dysphagia, dementia, depression, and complex socioeconomic influences, may also contribute to decreased food intake [5]. Finally, chronic inflammation plays a major role given its association with decreased appetite and food intake in older adults [6]. This is potentially due to the inflammation-driven loss of an appropriate hypothalamic response towards stimuli of appetite [7,8].

Since malnutrition is associated with immune dysfunction, muscle loss, and many other consequences, it is related to worse health-related outcomes, such as increased risks for infections, length of hospital stay, worse recovery after acute illness, and overall mortality [9]. The consequences of dehydration are similarly widespread. They include general weakness, decrease in cognitive function, and many others. Most importantly, dehydration is a risk factor for delirium, urinary tract infections, and cardiovascular disease, and it is a major cause for hospital admissions of older persons [10].

The sensation of thirst is induced via activation of osmoreceptors in response to increased extracellular osmolality during lack of body water [11]. Besides the initiation of voluntary drinking, secretion of the pituitary antidiuretic hormone arginine vasopressin (AVP) is induced to achieve renal water retention [11]. The mechanisms behind the often-diminished sensation of thirst in older adults [12,13] are not fully understood. Likely, it is due to a complex interplay between changes in the hormonal regulation of compensatory mechanisms coupled with a decline in renal function that reduces urinary concentrating ability [11]. In addition, the intake of commonly prescribed medications such as sodium glucose transporter (SGLT) 2 inhibitors [14] or loop diuretics can further increase the renal loss of body water and/or salt. Other contributing factors that reduce water intake at older age largely mirror those affecting food intake, such as dysphagia, dementia, and socioeconomic factors [11]. In addition, due to the high fluid content of fat-free mass, lower muscle mass is related to a higher risk of dehydration [11].

Dehydration in older adults is related to a variety of worse health-related outcomes, such as cognitive impairment [11], constipation [11], and increased risk for acute kidney failure [15] and delirium [16,17]. Given that dehydration is preventable, the early recognition of its signs and symptoms is crucial to mitigate these serious health-related consequences. As summarized above, the predominant type of dehydration in older adults is low-intake dehydration; thus, hyperosmolality is a valid marker to diagnose dehydration [18]. Based on that, the gold standard for diagnosing dehydration is the direct measurement of plasma osmolality [1,18]. If such a direct measurement is not available, it is recommended to calculate osmolality using serum concentrations of sodium, potassium, glucose, and urea [1,18]. If a cut-off of ≥295 mmol/L is used, a sensitivity of 85% and a specificity of 59% can be expected [19].

Given the detrimental impact of malnutrition and dehydration on health-related outcomes in older adults, their presumably shared risk factors, and considering the availability of potential preventive strategies, we aimed to investigate the overlap of malnutrition and dehydration as well as predisposing factors for dehydration in older hospitalized patients.

This study aims to investigate the overlap of malnutrition and dehydration and to identify predisposing factors for these conditions in older hospitalized patients. We hypothesize that malnutrition and dehydration, while commonly studied independently, share certain risk factors that could be addressed through integrated preventive strategies. This research seeks to clarify these relationships and contribute to improved clinical protocols for managing these prevalent issues in geriatric health.

## 2. Subjects and Methods

### 2.1. Respondents

This research is part of a larger multicenter, prospective, and observational study conducted from December 2022 to March 2024, aimed at assessing the causes of protein energy malnutrition among older adults, encompassing both inpatients and outpatients. It is important to note that the results of this larger study are not yet published. Within this extensive study, we additionally evaluated the laboratory characteristics indicative of dehydration of the participants in our department. A sample size calculation was performed for the primary aim of the study but not for this secondary analysis. Participants were recruited from three hospitals (n = 500), with the majority (n = 454) coming from Marien Hospital Herne and during routine clinical visits at a general practitioner’s office in Mülheim (Ruhr, n = 556).

Participants were included if they were aged 75 years or older and could provide an appropriate weight measurement. Exclusions were made for patients with terminal illness, acute fluid imbalance (such as decompensated heart failure or clinically obvious symptomatic dehydration), those requiring hemodialysis or parenteral nutrition for more than two weeks, and those with amputated limbs. However, for the purposes of the current study, only inpatient data were included since osmolality measurements, critical for assessing dehydration, were not available for outpatient participants.

Of the initial 500 inpatients, 46 were excluded due to the non-availability of osmolality measurements. Complete datasets were available for the remaining 454 inpatient participants (69% female and an age range of 75–99 years), and the analysis was conducted using this cohort.

### 2.2. Comprehensive Geriatric Assessment and Further Trial Measurements

All participants were evaluated by a comprehensive geriatric assessment (CGA) at the time of hospital admission assessing frailty, functional dependence, cognitive status, mood, nutritional status, and calf circumference. Furthermore, a routine laboratory assessment, including renal function tests and calculated serum osmolality tests, was performed.

Frailty was assessed with the Clinical Frailty Scale [20], which combines descriptive texts and pictograms to illustrate a patient’s functional and activity level, rated from 1 (very fit) to 9 (terminally ill). Functional dependence was assessed by the Barthel Index, which evaluates the patient’s ability to carry out self-care tasks. As such, 40–55/100 points correspond to a moderate functional dependence, whereas 20–35/100 points represent severe dependence and <20 points to a very severe functional dependence [21].

Cognitive function was assessed by the Montreal Cognitive Assessment (MoCA) test, as recently described [22]. Although a total score below 26/30 points assumes a cognitive impairment (depending on age, sex, and the level of education), there is no defined cut-off for moderate or severe dementia and the total score does not correlate explicitly with the severity of the cognitive impairment [22]. Despite that, we arbitrarily chose scores ≤ 20 points and ≤10 points to explore an association of cognitive dysfunction with malnutrition, as described below. The Depression in Old Age Scale (DiA-S) was performed to investigate depressive symptoms, and subjects were categorized as having no depression (0–2 points), suspected depression (3 point), and probable depression (4–10 points) [23].

### 2.3. Definition of Dehydration

Although several different types of dehydration do exist, this analysis focused on low-intake dehydration as the most common type in older adults [1].

Low-intake dehydration was identified using the following formula by the central laboratory:

Serum osmolality = 1.86 × serum sodium + serum glucose + serum urea + 9 [24].

Patients with a calculated osmolarity of ≥295 mmol/L were considered to be dehydrated [1,18]. Furthermore, it has to be noted that patients with clinically obvious and symptomatic severe dehydration were excluded from study participation, as this could potentially confound the weight-based criteria of malnutrition.

### 2.4. Assessment and Definition of Malnutrition

All patients were screened for malnutrition with the Mini Nutritional Assessment Short Form (MNA-SF) [25]. The MNA-SF has a total score of 14. Patients having scores of eight to eleven points were regarded as being at risk for malnutrition, with scores below eight points as manifested malnutrition. In addition, malnutrition was evaluated according to the Global Leadership Initiative on Malnutrition (GLIM) criteria. In brief, three phenotypic criteria included a non-voluntary weight loss of >5% within the past 6 months or >10% beyond 6 months, body mass index (BMI) < 22 kg/m^2^, and reduced muscle mass as measured by bioelectrical impedance (BIA), as well as two etiologic criteria of reduced food intake or assimilation as defined as <50% of energy requirements within the last week or any reduction for >2 weeks, any chronic gastrointestinal condition that adversely impacts food assimilation or absorption, and the presence of an inflammatory condition. Malnutrition was diagnosed if at least one phenotypic and one etiologic criterion was fulfilled [26,27]. Furthermore, patients were asked about the amount of weight loss along with the timeframe of this loss, categorizing the duration into three intervals of 3 months, 6 months, and >6 months.

### 2.5. Ethical Approval and Informed Consent

This study conforms to the principles laid down in the Declaration of Helsinki. Ethical approval from the local ethics committee at the Medical Faculty of Ruhr University (Bochum, Germany) was obtained before any data were collected (ethics vote no. 22-7629; date of approval: 14 October 2022). All participants gave their written informed consent before participating in the trial. The trial was registered in the German Clinical Trials Register (DRKS00030850).

### 2.6. Statistical Methods

The statistical analysis was performed with SPSS statistical software (SPSS Statistics for Windows, IBM Corp, Version 29.0, Armonk, NY, USA).

Continuous variables are reported with means and standard deviations (SDs) for normally distributed variables and median values with interquartile ranges (IQRs) for non-normally distributed data. Categorical variables are expressed as absolute numbers and relative frequencies (%).

Group comparisons were conducted using the t-test for continuous variables with a normal distribution, the Mann–Whitney U test for continuous variables with a non-normal distribution, and the Pearson chi-squared test for categorical variables. Additionally, binary logistic regression analysis was employed to investigate the impact of various risk factors, such as malnutrition, age, gender, BMI, previous weight loss, calf circumference, MoCA score, DiA-S score, Clinical Frailty Scale, Barthel Index, and creatinine (as independent variables) on dehydration (as the dependent variable). Moreover, the agreement between malnutrition and dehydration assessments was evaluated using the kappa statistic. A kappa value ranging from 0.80 to >0.90 was classified as high, a value from 0.60 to 0.79 as moderate, and a value from 0 to 0.59 as low [28]. A *p*-value of less than 0.05 was considered statistically significant.

## 3. Results

### 3.1. Characteristics of Study Participants

From the original database, 500 patients were categorized as inpatients. Of these, 46 were excluded due to missing values for osmolality. Complete datasets were available for the remaining 454 inpatient participants, and the analysis was conducted using this cohort. Patient characteristics are summarized in Table 1. Briefly, the mean age of participants was 82.21 years ± 6.82 (SD), and 69% of study participants were female. In total, 79% of the patients were classified as frail according to the Clinical Frailty Scale. The median score of the Barthel Index was 50 (IQR 40–60), suggesting moderate functional dependence. In addition, 17% (n = 78) of patients exhibited severe or very severe functional dependence, as defined by a Barthel Index ≤ 35 points. In terms of cognitive function, the median MoCA score was 18, with 72% (n = 308) of patients scoring ≤ 20 points and 9% (n = 40) scoring ≤ 10 points. Furthermore, 49% (n = 169) of the patients had probable depression according to the DiA-S.

### 3.2. Prevalence of Malnutrition and Dehydration

According to the GLIM criteria, 42% of participants (n = 191) were malnourished at the time of hospital admission. Additionally, using the MNA-SF, 45% of patients were classified as malnourished. The mean weight loss was 3.95 kg ± 6.32 (SD), with the majority of patients losing weight within the last 3 months prior to hospital admission compared to the last six months (29% vs. 7%). The mean calculated serum osmolality in the study cohort was 291.54 ± 10.40 mmol/L; 68% of participants (N = 307) were classified as normohydrated and 32% (N = 147) as dehydrated.

### 3.3. Comparison of Normohydrated and Dehydrated Patients

The comparison of health and nutritional status between normohydrated and dehydrated patients is depicted in Table 2. In the group of dehydrated patients, there was a significantly higher body weight, BMI, and baseline creatinine values, as well as a greater proportion of males. No statistically significant differences were observed in terms of frailty, cognitive function, positive screening for depression, or malnutrition, as assessed by either GLIM criteria or MNA-SF between both groups.

### 3.4. Overlap Between Malnutrition and Dehydration

The extent of overlap between dehydration and malnutrition according to GLIM criteria is illustrated in a Venn diagram depicted in Figure 1. Of the total study population, 13% (n = 57) of patients exhibited both malnutrition and dehydration at the time of hospital admission. The kappa statistic result was −0.045, indicating no agreement beyond chance between malnutrition and dehydration. Using the MNA-SF, similar patterns were observed but with slightly different values; the overlap was 14% (n = 65).

As cognitive impairment is a known factor associated with malnutrition, we assessed a possible association in our cohort. An MoCA score ≤ 20 points was significantly associated with an MNA-SF score of <8 points (*p* < 0.001). When malnutrition was defined according to GLIM criteria, an MoCA score ≤ 20 points was not significantly associated with malnutrition (*p* = 0.51).

To assess the independent effects of various risk factors on dehydration, such as malnutrition according to GLIM criteria, age, gender, BMI, previous weight loss, calf circumference, MoCA score, DiA-S score, Clinical Frailty Scale, Barthel Index, and creatinine, we conducted a binary logistic regression analysis. The results are presented in Table 3. Of all the factors included, only BMI and creatinine levels were found to be significantly associated with dehydration risk. None of the other variables demonstrated a statistically significant impact on dehydration. Additionally, when using malnutrition assessments based on the MNA-SF, we found similar results albeit with different values.

## 4. Discussion

In this exploratory analysis of our previous larger study, we examined the co-occurrence of dehydration and malnutrition. Remarkably, only 13% of patients showed an overlap of both conditions compared to 42% presenting with malnutrition and 32% with dehydration alone. Predictive factors for dehydration were notably a higher BMI and increased creatinine levels, whereas cognitive and mental status did not show a significant impact. To our knowledge, this represents the largest analysis of this topic in an inpatient cohort.

Nearly one-third of our patients were dehydrated at hospital admission. This rate aligns with previous studies. For instance, in a prospective cohort study among older adults acutely admitted to a large teaching hospital in Great Britain, 37% of patients ≥ 65 years were found to be dehydrated [29]. Of note, the prevalence of dehydration might have been underestimated in our study, as patients with severe symptomatic dehydration were excluded from trial participation to avoid a bias for weight-based nutritional assessment.

In our cohort, the prevalence of malnutrition, as assessed by MNA-SF, was as high as 45%, which is comparable to those described in other German [30], Australian [31], and international [32] inpatient geriatric populations. However, prevalence rates can vary significantly depending on the patient population [32] and the applied assessment method. In summary, the prevalence rates of malnutrition and dehydration which were observed in our study are considered representative and likely do not appear to introduce a significant bias.

The co-occurrence of malnutrition and dehydration has not been as thoroughly investigated as their individual prevalences. Bech and colleagues [4] prospectively analyzed 114 patients ≥65 years admitted to a geriatric ward. Of these, 49 (43%) were dehydrated, and a co-occurrence of dehydration and malnutrition was found in 30/114 (26%) patients when malnutrition was classified according to GLIM criteria. Our patient cohort had very similar inclusion and exclusion criteria, with the exception that patients with obvious and clinically symptomatic dehydration were excluded. This could potentially explain the lower co-occurrence we observed, as we might have missed severe cases. In addition, Bech and colleagues described a statistically significant association of dehydration with higher weight, height, mid-upper arm circumference, male gender, estimated glomerular filtration rate (eGFR) < 30 mL/min, and diabetes compared to normohydration [4]. Notably, several factors, such as weight, male gender, and impaired kidney function are in line with our findings.

In another analysis of an outpatient cohort comprising 1409 patients ≥ 60 years, the prevalence of dehydration, determined by calculated serum osmolality, was 33%, while the prevalence of malnutrition, assessed by MNA, was 19%. The co-occurrence of malnutrition and dehydration was 18% among dehydrated patients and 30% among those with malnutrition [33]. Of note, factors including polypharmacy, hypertension, diabetes mellitus, and higher BMI were associated with dehydration. In contrast, older age, a lower level of education, depression, and lower BMI were associated with malnutrition [33]. As already briefly discussed above, it is important to discuss that severe and clinically obvious dehydration led to the exclusion of a patient in our cohort, potentially omitting some patients with acute and severe illnesses or advanced dementia whose oral intake is inherently reduced. This might explain why the prevalence of dehydration is very similar, although an inpatient cohort usually includes more patients who are critically ill (and thus dehydrated). In addition, the prevalence of malnutrition was lower in the outpatient cohort than in our inpatient participants, which is as expected, and they often display features of severe illnesses.

Another trial assessed adults ≥ 65 years at an internal medicine outpatient clinic in Japan for hypertonic dehydration (defined as calculated serum osmolality ≥ 300 mos/kg) [34]. In that setting, the prevalence of hypertonic dehydration was 30.8%. As significant predictive factors, BMI ≥ 25 kg/m^2^, polypharmacy with daily intake of more than ten medications per day, and a daily fluid intake per body weight < 20 mL/kg/day were found [34].

As BMI/higher weight appeared as associated factors for dehydration across multiple cohorts [4,33,34], including ours, the exploration of potential pathophysiological mechanisms underlying this observation is pertinent. Obesity is associated with chronic kidney disease [35], which however does not explain a direct impact on dehydration. Potential explanations could be as follows: Body water is mainly stored within the fat-free mass (FFM), including the muscular compartment. At a higher BMI, particularly in older adults, there is often an increased prevalence of sarcopenic obesity, a condition that is defined by a loss of muscle mass and function together with features of obesity [36]. Thus, a higher BMI can potentially go along with a reduction in storage capacity for body water in a low FFM, particularly if measured per kilogram of body weight [37]. In addition, obesity is associated with a larger body surface area and altered body heat production and emission, which may result in increased sweating and subsequent dehydration. Furthermore, the altered body composition that often accompanies aging could disrupt the normal hormonal and inflammatory responses that regulate fluid balance, exacerbating the dehydration risk [37]. This complex interplay of factors might explain our findings, linking higher BMI and weight with increased dehydration risk in our and other study cohorts.

The strong association between elevated serum creatinine and dehydration is expected, as increased creatinine levels often result from dehydration. Elevated creatinine is typically more a consequence than a risk factor for dehydration. Therefore, individuals with higher serum creatinine should be routinely screened for dehydration.

In our cohort, we found no association of impaired cognitive function and dehydration; for malnutrition, a significant association was present only if malnutrition was defined based on MNSA-SF. This is somehow surprising, as dementia was described as a risk factor for malnutrition [5,38] and dehydration [11], respectively. A possible and likely explanation is a potential bias in our study cohort. As compared to a retrospective analysis of electronic medical records including all patients admitted to the geriatric ward, trial participation required informed consent by the patients or their authorized representatives if patients did not maintain their legal capacity. The latter is usually related to greater efforts and time consumption in study recruitment and the subsequent risk that patients with dementia are systematically excluded. Thus, patients with severe dementia who forget to eat and drink might have been underrepresented. This would explain that all patients were able to actively participate in the cognitive testing, which requires a reasonable extent of skills. In addition, oropharyngeal dysphagia predisposes to a co-occurrence of malnutrition and dehydration [39]. Dysphagia is common in patients with advanced dementia [40]; thus, this specific risk group might have been missed in our analysis.

The regulation of hunger and thirst involves different underlying mechanisms. While malnutrition and dehydration, both outcomes of impaired sensations and homoeostasis, may share common risk factors, their separate appearance might follow an evolutionary shaped pattern.

Thirst regulation has an evolutionary conserved mechanism [41]. For instance, in birds and terrestrial mammals, a rise in plasma osmolality triggers a dipsogenic stimulus [41] very similar to the main regulatory pathways in humans. However, thirst regulation in humans appears to have a greater complexity and a multifactorial regulation. As a clinical example, although thirst is frequently observed in end-of-life situations in humans, it does not always correlate with increased osmolality in such cases [42,43].

On the other hand, the tolerance of dehydration within a defined safety margin appears to be evolutionary, inevitably required to hunt or seek water during the heat of the day without significant performance deficits [44]. Thus, it has been suggested that cognitive function can be maintained during dehydration if it occurs during physical exercise, at least in younger adults [45]. This window of tolerance for increased osmolality may be relevant even nowadays in patients with chronic heart failure who are under efficient treatments. This suggests that not all patients classified as dehydrated in our cohort face a condition that necessarily requires correction, particularly regarding the low osmolality threshold for the categorization of dehydration in our analysis. Furthermore, we would expect that symptomatic dehydration occurs mainly if the capacity to compensate for fluid depletion, i.e., anti-diuresis, decrease in sweating, etc., is exhausted or itself is leading to relevant symptoms. Factors such as changes in body composition (e.g., higher BMI, criteria for malnutrition, likely reduced muscle mass) that result in inadequate water storage, together with diminished renal function affecting urine concentration, may lead to an impaired compensatory capacity for dehydration. This could be the reason why both factors were identified as significant contributors to dehydration in our cohort.

The minimal co-occurrence of malnutrition and dehydration in our study underscores the importance of separate screening and assessment strategies. While both conditions are frequent in older persons and fluid intake is part of nutritional intake, integrating nutritional screening and assessment tools that may simultaneously predict or detect dehydration would be particularly valuable in resource-limited settings such as general practices and nursing homes [46]. This approach aligns with ESPEN guidelines, which advocate for proactive hydration management [1]. Enhancing training for clinical staff to recognize the distinct and subtle signs of each condition can facilitate early detection and appropriate intervention. Furthermore, the distinct pathways leading to malnutrition and dehydration highlight the need for tailored interventions. Additionally, the urgent need for more research into the mechanisms of low-intake dehydration should be prioritized to better understand and address these challenges.

This study has several limitations that must be acknowledged. Firstly, the cross-sectional nature of our analysis prevents us from establishing causal relationships between dehydration, malnutrition, risk factors, and health-related outcomes. Longitudinal studies would be required to determine the directionality of these associations. Secondly, the likely underrepresentation of patients with severe dehydration, severe dementia, or end-stage diseases may limit the generalizability of our findings to all different populations of older adults. This exclusion might have resulted in an underestimation of the co-occurrence of malnutrition and dehydration, which are often prevalent in these patient groups based on our clinical experience. Thirdly, the use of calculated serum osmolality as a proxy for dehydration might have limitations. A direct measurement of osmolality would have been preferable but was not feasible in this study.

## 5. Conclusions

Our study confirms that malnutrition and dehydration are distinct pathophysiological conditions in older adults and do not co-occur as frequently as might be expected. In addition, we identified that BMI and creatinine levels are significant predictors of dehydration risk among this population. Consequently, the implementation of separate screening and assessment strategies for malnutrition and dehydration is recommended to better identify and address these conditions individually.

## Figures and Tables

**Figure 1 nutrients-17-01004-f001:**
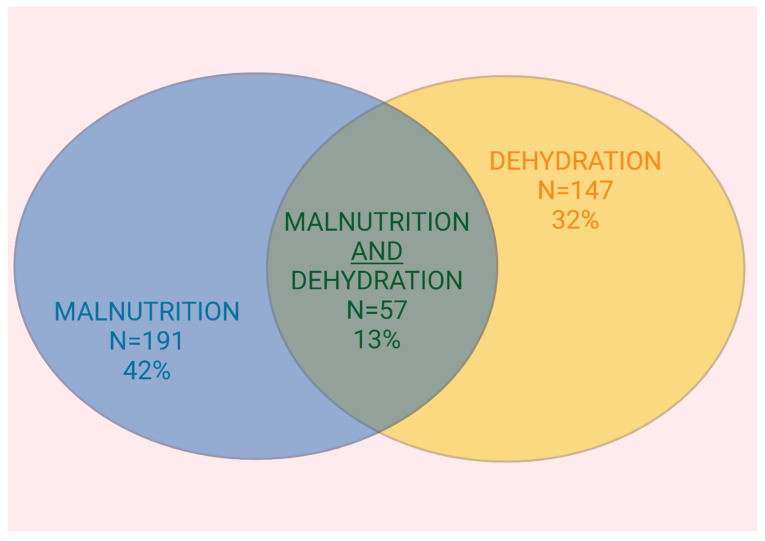
Overlap between malnutrition and dehydration. Legend: Venn diagram displaying the extent of overlap between dehydration and malnutrition according to GLIM criteria. The percentages displayed in the diagram are calculated based on the total population studied.

**Table 1 nutrients-17-01004-t001:** Characteristics of study population upon hospital admission.

	All (n = 454)
Gender	
Female, n (%)	315 (69)
Male, n (%)	139 (31)
Age (y), mean ± SD	82.21 ± 6.82
Nutritional status	
Height (m), mean ± SD	1.65 ± 0.09
Body weight (kg), mean ± SD	72.70 ± 17.83
BMI (kg/m^2^), mean ± SD	26.50 ± 6.02
Calf circumference (cm), mean ± SD	33.63 ± 4.82
Previous weight loss, (kg), mean ± SD	3.95 ± 6.32
Duration of previous weight loss (month), mean ± SD	
<3 months	131 (30)
<6 months	30 (7)
GLIM criteria for the diagnosis of malnutrition	
Malnourished, n (%)	191 (42)
Not malnourished, n (%)	263 (58)
MNA-SF, median (IQR)	8 (6–9)
Normal nutritional status, n (%)	20 (5)
Risk of malnutrition, n (%)	230 (50)
Malnourished, n (%)	204 (45)
Geriatric assessment	
Barthel Index, median (IQR)	50 (40–60)
MoCa, median (IQR)	18 (15–21)
DiA-S, median (IQR)	3 (1–5)
Clinical Frailty Scale, median (IQR)	5 (5–6)
Frailty, n (%)	357 (79)
No frailty, n (%)	97 (21)
Osmolality (mmol/L), mean ± SD	291.54 ± 10.40
Hydration status	
Normohydrated, n (%)	307 (68)
Dehydrated, n (%)	147 (32)
Creatinine (mg/dL), mean ± SD	1.11 ± 0.54

Abbreviations: MNA-SF, Mini Nutritional Assessment Short Form; MoCa, Montreal Cognitive Assessment; DiA-S, Depression in Old Age Scale. Values are given as numbers (%), mean ± SD (standard deviation), or median (IQR, interquartile range).

**Table 2 nutrients-17-01004-t002:** Group comparison between normohydrated and dehydrated patients.

	Normohydrated (n = 307, 68%)	Dehydrated (n = 147, 32%)	*p* Value
Gender			
Female, n (%)	225 (71)	90 (29)	0.012
Male, n (%)	82 (59)	57 (41)	
Age (y), mean ± SD	81.91 ± 6.90	82.93 ± 6.65	0.147
Height (m), mean ± SD	1.65 ± 0.08	1.66 ± 0.09	0.191
Body weight (kg), mean ± SD	70.93 ± 16.93	76.30 ± 19.04	0.004
BMI (kg/m^2^), mean ± SD	26.0 ± 5.68	27.61 ± 6.51	0.010
Calf circumference (cm), mean ± SD	33.50 ± 4.72	33.92 ± 4.91	0.397
Previous weight loss, (kg), mean ± SD	3.87 ± 6.30	4.03 ± 6.52	0.772
Duration of previous weight loss (month), mean ± SD			
<3 months	92 (30)	39 (27)	0.533
<6 months	17 (5)	13 (9)	
GLIM criteria for the diagnosis of malnutrition			
Malnourished, n (%)	134 (44)	57 (39)	0.361
Not malnourished, n (%)	173 (56)	90 (61)	
Geriatric assessment			
MNA-SF, median (IQR)	8 (6–9)	8 (6–9)	0.991
Normal nutritional status, n (%)	12 (4)	8 (6)	0.756
Risk of malnutrition, n (%)	156 (51)	74 (50)	
Malnourished, n (%)	139 (45)	65 (44)	
Barthel Index, median (IQR)	50 (40–60)	50 (40–60)	0.357
MoCa, median (IQR)	18 (15–21)	18 (15–21)	0.371
DiA-S, median (IQR)	3 (1–5)	3 (1–5)	0.270
Clinical Frailty Scale, median (IQR)	5 (5–6)	5 (5–6)	0.093
Frailty, n (%)	240 (78)	117 (80)	0.807
No frailty, n (%)	67 (22)	30 (20)	
Osmolality (mOsm/kg), mean ± SD	286.44 ± 8.05	302.01 ± 6.00	<0.001
Creatinine (mg/dL), mean ± SD	0.92 ± 0.41	1.41 ± 0.64	<0.001

Abbreviations: BMI, body mass index; DiA-S, Depression in Old Age Scale; IQR, interquartile range; n, number; SD, standard deviation.

**Table 3 nutrients-17-01004-t003:** Binary regression analysis of risk factors associated with dehydration.

	Dehydration (Yes/No)					
				95% CI for Exp(B)		
	B	SE	Exp(B)	Lower	Upper	*p* Value
Malnutrition (yes/no)	0.040	0.330	1.041	0.545	1.989	0.903
Age (year)	0.008	0.019	1.008	0.971	1.047	0.661
Gender (female/male)	−0.063	0.273	0.939	0.550	1.603	0.816
BMI (kg)	0.072	0.031	1.075	1.011	1.142	0.020
Previous weight loss (kg)	0.024	0.024	1.024	0.976	1.073	0.331
Calf circumference (cm)	−0.052	0.037	0.949	0.883	1.020	0.158
MoCA score	−0.018	0.026	0.982	0.933	1.034	0.489
DiA-S score	0.009	0.051	1.009	0.914	1.114	0.857
CSF (points)	0.078	0.102	1.081	0.886	1.319	0.444
Barthel Index	0.001	0.008	1.001	0.985	1.017	0.892
Creatinine (mg/dL)	2.185	0.336	8.889	4.601	17.174	<0.001

Abbreviations: BMI, body mass index; CI, confidence interval; CSF, Clinical frailty scale; SE, standard error.

## Data Availability

The data presented in this study are available on request from the corresponding author. The data are not publicly available due to ongoing analysis of the study.

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
