# Peer review of "Dehydration and Malnutrition—Similar Yet Different: Data from a Prospective Observational Study in Older Hospitalized Patients"

_nutrients, 2025, doi:10.3390/nu17061004_

Round 1
Reviewer 1 Report
Comments and Suggestions for Authors
I find the article clear and complete. However, to enhance reader interest, I suggest adding information on the recommended caloric intake and key nutrients to consider in the diet in in older hospitalized patients ans in healthy subjects. This could be presented in a table, including Substance (e.g., Carnitine), Suggested Dosage, Importance/Benefits, Reference (e.g., DOI: 10.3390/jfmk6040093)
Author Response
ANSWERS TO REVIEWERS
We thank the referee for her/his interest in our work and for helpful comments that improved the manuscript. We have tried to do our best to respond to the points raised. The referee has brought up some good points and we appreciate the opportunity to clarify our research objectives and results. As indicated below, we have checked all the general and specific comments provided by the referee and have made necessary changes (highlighted in yellow throughout the text) according to her/his indications.
Reviewer 1
I find the article clear and complete. However, to enhance reader interest, I suggest adding information on the recommended caloric intake and key nutrients to consider in the diet in in older hospitalized patients and in healthy subjects. This could be presented in a table, including Substance (e.g., Carnitine), Suggested Dosage, Importance/Benefits, Reference (e.g., DOI: 10.3390/jfmk6040093)
Thank you for your suggestion to include information on recommended caloric intake and key nutrients such as Carnitine. We appreciate your interest in broadening the scope of the article. However, our study specifically aimed to investigate the relationship between malnutrition and dehydration in older hospitalized patients, with a focus on clinical outcomes related to these conditions. Including detailed nutritional guidance, including specifics on substances like Carnitine, was beyond the primary scope of our manuscript. We believe that maintaining a focused approach ensures clarity and relevance to the study's objectives.
Reviewer 2 Report
Comments and Suggestions for Authors
I have completed the review of the article entitled „Dehydration and malnutrition – similar but yet different: data from a prospective observational study in older hospitalized patients“ and thank you for the opportunity to contribute to the evaluation of this research. I hope that my comments and suggestions will be useful for the improvement of the article.
The introduction is informative and provides basic information on the topic. Nevertheless, the hypotheses and the aim of the study, which are crucial for the review and analysis, are missing, while the aim defines the direction and purpose of the article. Without these elements, the study is unclear in terms of expected outcomes and contributions.
The chapter “Purpose of the study” should be titled “Respondents” to make it clear that this is a description of the research participants. This section lacks a detailed description of the respondents, including the recruitment of participants and the inclusion and exclusion criteria. The response rate is also not provided, which is key information for assessing the representativeness of the sample. In addition, the timing of the study (start and end) is not specified.
The authors mention that the study is part of a larger cross-sectional and longitudinal study, but do not provide a single reference to confirm this. This lack of references reduces the transparency and credibility of the study. The description of the subjects should be much more detailed and precise so that readers can better understand the sample and methodology.
With regard to the CONSORT diagram, the question arises as to why outpatients (N=556) were included if they were immediately excluded from the study. Based on the available information, these patients appear to have been excluded because they had normal osmolarity or this not available, which makes them outpatients. The authors state: “However, for the purposes of this exploratory analysis, only in-patient data were included as osmolality measurements were not available for outpatient participants.” This statement is somewhat unclear and requires further clarification. It is not clear why osmolarity data was only available for inpatients and not for outpatients. If so, then they did not meet the inclusion criteria. It is not clear why these patients are included here, as they were not the target group. Authors should define and describe terminally ill patients in the methodology, since they mention them in the conclusion, but not here. In addition, the flowchart should be clearer and show each step of inclusion and exclusion of respondents to ensure transparency and reproducibility of the research. This would avoid ambiguity and provide a better understanding of the recruitment process and sample characteristics.
In addition, the scales used to assess should be described in more detail For example, Depression in the Elderly scale, the structure of the scale, the range of scores and the criteria for diagnosing depression should be explained. It should be stated what the scales are, how they are scored, what the minimum and maximum scores are and how the results are to be interpreted. In general, chapter on respondents needs to be significantly improved to make it clearer, more transparent and more informative. A more detailed description of the methodology, the instruments used and the characteristics of the sample is essential for a critical evaluation and replication of the study.
The discussion is well developed and provides a detailed analysis of the results obtained, which are placed in the context of the existing literature. The authors have successfully linked their findings to previous research, highlighting similarities and differences, which contributes to a better understanding of the research problem. It is necessary to mention the implications for clinical practice and future research. Overall, the discussion is well structured, argumentative and informative, making it one of the strongest parts of the thesis.
The conclusion is inadequate as it does not emphasize the main findings of the study, particularly the relationship between body mass index (BMI) and dehydration, which is the main contribution of this study. Instead, the conclusion is too general and does not provide a clear synthesis of the main findings. Furthermore, it is not clear what needs to be done to address the issue discussed. According to this, conclusion needs to be imporoved, focusing on main findings, especially those related to BMI and dehydration. It is not necessary to mention the data on terminally ill patients in the conclusion. This is a further indication that the chapter on respondents and the flowchart should be clearly supplemented and elaborated. These patients were not described in detail in the methodology nor included in the flowchart. The methodology needs to clearly define who terminally ill patients are, what criteria were used for their inclusion or exclusion and how they influenced the results of the study. If these patients were not part of the main analysis, their mention in the conclusion may confuse readers and reduce the clarity of the statement. The conclusion should be focused on the main findings of the study, such as the relationship between BMI and dehydration, while terminally ill patients, should either be explained in detail in the methodology or omitted from the conclusion altogether.
Author Response
ANSWERS TO REVIEWERS
We thank the referee for her/his interest in our work and for helpful comments that improved the manuscript. We have tried to do our best to respond to the points raised. The referee has brought up some good points and we appreciate the opportunity to clarify our research objectives and results. As indicated below, we have checked all the general and specific comments provided by the referee and have made necessary changes (highlighted in yellow throughout the text) according to her/his indications.
Reviewer 2
I have completed the review of the article entitled „Dehydration and malnutrition – similar but yet different: data from a prospective observational study in older hospitalized patients“ and thank you for the opportunity to contribute to the evaluation of this research. I hope that my comments and suggestions will be useful for the improvement of the article.
The introduction is informative and provides basic information on the topic. Nevertheless, the hypotheses and the aim of the study, which are crucial for the review and analysis, are missing, while the aim defines the direction and purpose of the article. Without these elements, the study is unclear in terms of expected outcomes and contributions.
Thank you for your suggestion. We have updated the introduction to clearly define the study's aims and hypotheses, ensuring the objectives and expected outcomes are explicitly stated, please see page 3 lines 97-102.
The chapter “Purpose of the study” should be titled “Respondents” to make it clear that this is a description of the research participants. This section lacks a detailed description of the respondents, including the recruitment of participants and the inclusion and exclusion criteria. The response rate is also not provided, which is key information for assessing the representativeness of the sample. In addition, the timing of the study (start and end) is not specified.
Thank you for your comment. We have clarified the inclusion and exclusion criteria, recruitment information, the response rate and the timing of the study in the manuscript text to provide a comprehensive understanding of the study's scope and timeframe. Please see page 3 lines 105-124.
The authors mention that the study is part of a larger cross-sectional and longitudinal study, but do not provide a single reference to confirm this. This lack of references reduces the transparency and credibility of the study. The description of the subjects should be much more detailed and precise so that readers can better understand the sample and methodology.
The results of the larger study that this research is part of, have not yet been published, hence the absence of a reference. This study serves as a secondary analysis focused specifically on dehydration, using data originally collected to explore protein-energy malnutrition among older adults, which included both inpatients and outpatients. It has been mentioned in lines 105-113.
With regard to the CONSORT diagram, the question arises as to why outpatients (N=556) were included if they were immediately excluded from the study. Based on the available information, these patients appear to have been excluded because they had normal osmolarity or this not available, which makes them outpatients. The authors state: “However, for the purposes of this exploratory analysis, only in-patient data were included as osmolality measurements were not available for outpatient participants.” This statement is somewhat unclear and requires further clarification. It is not clear why osmolarity data was only available for inpatients and not for outpatients. If so, then they did not meet the inclusion criteria. It is not clear why these patients are included here, as they were not the target group. Authors should define and describe terminally ill patients in the methodology, since they mention them in the conclusion, but not here. In addition, the flowchart should be clearer and show each step of inclusion and exclusion of respondents to ensure transparency and reproducibility of the research. This would avoid ambiguity and provide a better understanding of the recruitment process and sample characteristics.
Thank you for your comment. This study is a secondary analysis of data originally collected to assess the causes of protein-energy malnutrition among older adults, which included both inpatients (n = 500) and outpatients from general practitioner offices (n = 556). While hydration was also monitored, it was not the primary focus of the initial study. Limited data availability from GP practices, particularly regarding osmolality measurements, necessitated their exclusion from this specific analysis focused on hydration. This exclusion was due to the inability to obtain comprehensive hydration-related data in outpatient settings compared to inpatient settings, where such data were readily available. We have clarified this process more clearly in the text, lines 96-117.
We have omitted the flowchart (Figure 1) to prevent redundancy and have instead incorporated detailed descriptions directly into the text to ensure clarity and transparency regarding the recruitment process and sample characteristics.
In addition, the scales used to assess should be described in more detail For example, Depression in the Elderly scale, the structure of the scale, the range of scores and the criteria for diagnosing depression should be explained. It should be stated what the scales are, how they are scored, what the minimum and maximum scores are and how the results are to be interpreted. In general, chapter on respondents needs to be significantly improved to make it clearer, more transparent and more informative. A more detailed description of the methodology, the instruments used and the characteristics of the sample is essential for a critical evaluation and replication of the study.
Thanks for your suggestion. We have added the scale used to assess the geriatric assessment. Please see page 3 and 4 section: Comprehensive geriatric assessment and further trial measurements.
The discussion is well developed and provides a detailed analysis of the results obtained, which are placed in the context of the existing literature. The authors have successfully linked their findings to previous research, highlighting similarities and differences, which contributes to a better understanding of the research problem. It is necessary to mention the implications for clinical practice and future research. Overall, the discussion is well structured, argumentative and informative, making it one of the strongest parts of the thesis.
Thank you for your positive feedback. Based on your suggestion, we have emphasized the implications for clinical practice and future research in page 10 lines 378-388.
The conclusion is inadequate as it does not emphasize the main findings of the study, particularly the relationship between body mass index (BMI) and dehydration, which is the main contribution of this study. Instead, the conclusion is too general and does not provide a clear synthesis of the main findings. Furthermore, it is not clear what needs to be done to address the issue discussed. According to this, conclusion needs to be imporoved, focusing on main findings, especially those related to BMI and dehydration. It is not necessary to mention the data on terminally ill patients in the conclusion. This is a further indication that the chapter on respondents and the flowchart should be clearly supplemented and elaborated. These patients were not described in detail in the methodology nor included in the flowchart. The methodology needs to clearly define who terminally ill patients are, what criteria were used for their inclusion or exclusion and how they influenced the results of the study. If these patients were not part of the main analysis, their mention in the conclusion may confuse readers and reduce the clarity of the statement. The conclusion should be focused on the main findings of the study, such as the relationship between BMI and dehydration, while terminally ill patients, should either be explained in detail in the methodology or omitted from the conclusion altogether.
Thank you for your feedback regarding the conclusion of our study. We have revised the conclusion to emphasize the main findings more clearly, particularly the significant association between BMI and dehydration risk. Please see page 10, section conclusion as well as abstract.
Reviewer 3 Report
Comments and Suggestions for Authors
I think this is a very valuable and thorough study on the differences between dehydration and malnutrition in the elderly. Here are some of the excellent points of this paper:
- Clear Objectives: The objective of exploring the association between malnutrition and dehydration in older hospitalized patients is clearly stated.
- Concise Methods: The description of the study design, assessments used (MNA-SF, GLIM, serum osmolality), and statistical methods is brief but sufficient.
- Key Results Highlighted: The prevalence of malnutrition and dehydration, the low overlap between the two, and the identified predictive factors for dehydration are clearly presented.
- Logical Conclusion: The conclusion that malnutrition and dehydration are common but don't frequently co-occur, and the explanation based on different regulatory pathways, is well-supported by the results,
Although the authors have mentioned the points below, I think it would be easier for readers to understand if they had included more of these points.
- Defining Dehydration: While serum osmolality is mentioned, the specific cutoff value used to define dehydration should be explicitly stated. This is crucial for reproducibility and interpretation of the results.
- Sample Size and Demographics: While the total number of patients is provided (454), some basic demographic information (e.g., age, sex) would be helpful to understand the study population better. Were there specific inclusion/exclusion criteria?
- Clinical Significance of Predictive Factors: While elevated serum creatinine and higher BMI were identified as predictive factors for dehydration, the clinical significance of these findings could be elaborated upon. Are these factors readily modifiable? What are the implications for clinical practice?
- Limitations: The summary would benefit from briefly acknowledging any limitations of the study. For example, it's an exploratory analysis, so causal relationships can't be established. Were there any limitations in data collection or potential for bias? The use of calculated serum osmolality as a proxy for dehydration might be discussed as potentially having limitations. Direct measures of hydration status might have been preferable, if feasible.
- "Scarce Data" Claim: The introduction mentions "scarce data" on the co-occurrence of malnutrition and dehydration. While this may be true, it would be strengthened by citing a few relevant studies (or lack thereof) to support this claim.
- Mechanism Discussion: The conclusion mentions different regulatory pathways for thirst and hunger. Briefly elaborating on these pathways (e.g., hormonal influences, neurological control) could add depth to the explanation.
Author Response
ANSWERS TO REVIEWERS
We thank the referee for her/his interest in our work and for helpful comments that improved the manuscript. We have tried to do our best to respond to the points raised. The referee has brought up some good points and we appreciate the opportunity to clarify our research objectives and results. As indicated below, we have checked all the general and specific comments provided by the referee and have made necessary changes (highlighted in yellow throughout the text) according to her/his indications.
Reviewer 3
I think this is a very valuable and thorough study on the differences between dehydration and malnutrition in the elderly. Here are some of the excellent points of this paper:
Clear Objectives: The objective of exploring the association between malnutrition and dehydration in older hospitalized patients is clearly stated.
Concise Methods: The description of the study design, assessments used (MNA-SF, GLIM, serum osmolality), and statistical methods is brief but sufficient.
Key Results Highlighted: The prevalence of malnutrition and dehydration, the low overlap between the two, and the identified predictive factors for dehydration are clearly presented.
Logical Conclusion: The conclusion that malnutrition and dehydration are common but don't frequently co-occur, and the explanation based on different regulatory pathways, is well-supported by the results,
Thank you for your encouraging feedback on our study.
Although the authors have mentioned the points below, I think it would be easier for readers to understand if they had included more of these points.
Defining Dehydration: While serum osmolality is mentioned, the specific cutoff value used to define dehydration should be explicitly stated. This is crucial for reproducibility and interpretation of the results.
Thanks for your suggestion. We have edited this part. Please see lines page 4, 150-153.
Sample Size and Demographics: While the total number of patients is provided (454), some basic demographic information (e.g., age, sex) would be helpful to understand the study population better. Were there specific inclusion/exclusion criteria?
Thank you for your recommendation. We have thoroughly outlined detailed demographic information, along with inclusion and exclusion criteria in the text (please see page 3, lines 105-125).
We have omitted the flowchart (Figure 1) to prevent redundancy and have instead incorporated detailed descriptions directly into the text to ensure clarity and transparency regarding the recruitment process and sample characteristics.
Clinical Significance of Predictive Factors: While elevated serum creatinine and higher BMI were identified as predictive factors for dehydration, the clinical significance of these findings could be elaborated upon. Are these factors readily modifiable? What are the implications for clinical practice?
Thank you for your comment. BMI is indeed a modifiable factor, and interventions aimed at weight management can significantly impact an individual's overall health and hydration status. Elevated serum creatinine, a marker of kidney function, can also be addressed with appropriate medical treatment to improve kidney health and hydration status in at-risk populations. However, an increase of creatinine is a typical consequence of dehydration and is therefore statistically associated with dehydration. We have mentioned in our discussion, page 9 lines 333-336.
Limitations: The summary would benefit from briefly acknowledging any limitations of the study. For example, it's an exploratory analysis, so causal relationships can't be established. Were there any limitations in data collection or potential for bias? The use of calculated serum osmolality as a proxy for dehydration might be discussed as potentially having limitations. Direct measures of hydration status might have been preferable, if feasible.
Thanks for your suggestion. We have addressed this concern in the limitations section of our manuscript, please see page 11 lines 398-400.
"Scarce Data" Claim: The introduction mentions "scarce data" on the co-occurrence of malnutrition and dehydration. While this may be true, it would be strengthened by citing a few relevant studies (or lack thereof) to support this claim.
Thank you for your suggestion. We have discussed this topic in more detail in the discussion section of our manuscript (lines 297-315). However, based on your feedback, we have also included a brief mention in the introduction to further support our statement about the scarcity of data. This addition can be found on lines 43-46 of the manuscript.
Mechanism Discussion: The conclusion mentions different regulatory pathways for thirst and hunger. Briefly elaborating on these pathways (e.g., hormonal influences, neurological control) could add depth to the explanation.
Thank you for your suggestion. We have already revised the conclusion based on other feedback. However, for a detailed discussion on the mechanisms, please refer to lines 352-360 in the discussion section, where these topics are thoroughly explored.
Reviewer 4 Report
Comments and Suggestions for Authors
Dear authors,
This study was conducted to dehydration and malnutrition – similar but yet different: data from a prospective observational study in older hospitalized patients. This manuscript has been well designed, written, and systematic. Moreover, it was a very good topic for older hospitalized patients. I believe this study was excellent issue in field of nutrition, medicine, and public health section.
Abstract:
Please add all results of statistical exact p-value in each variable in Results section
Line 35: please sort alphabetically in Key-words.
Introduction:
Please reformat your entire manuscript to adhere to the MDPI journal's guidelines. For example, Line 39: ‘older adults.1 Anorexia of aging2’ to ‘older adults [1]. Anorexia of aging [2]’.
Please, you should more explain the literature reviews and backgrounds between dehydration and malnutrition factors. For example, you should add immune, inflammation, or nutrition point of view. I recommend that you should be added 4-5 paragraphs for backgrounds.
Method: Well-written
In Figure 1, please change the beginning and first letter of all words to uppercase.
Line 127: please delete whole sentence of line 127.
‘Nutritional assessment is described in a later section in greater detail.’
Results: Well-written
Line 183: Please delete sentence of ‘The CONSORT diagram is depicted in Figure 1.’. You already describe in Methods section.
Line 172: DiA-S, Line 191: DIA-S. Please unify the terminology in whole manuscript.
In Table 1: Please, the full name and abbreviation of SD in abbreviations section.
Line 194 and 212: Change ‘DiA-S, Depression in old Age’ to ‘DiA-S, depression in old age scale’, Please unify the terminology, too
You may revise all results to two decimal places in mean, standard deviation, etc., and three decimal places in statistical values (t, F value, p-value) are generally spelled out in academic writing. Please change in whole results and all Tables.
Discussion
- You should add more applications in this study in Discussion section.
Author Response
ANSWERS TO REVIEWERS
We thank the referee for her/his interest in our work and for helpful comments that improved the manuscript. We have tried to do our best to respond to the points raised. The referee has brought up some good points and we appreciate the opportunity to clarify our research objectives and results. As indicated below, we have checked all the general and specific comments provided by the referee and have made necessary changes (highlighted in yellow throughout the text) according to her/his indications.
Reviewer 4
Dear authors,
This study was conducted to dehydration and malnutrition – similar but yet different: data from a prospective observational study in older hospitalized patients. This manuscript has been well designed, written, and systematic. Moreover, it was a very good topic for older hospitalized patients. I believe this study was excellent issue in field of nutrition, medicine, and public health section.
Abstract:
Please add all results of statistical exact p-value in each variable in Results section
Thanks for your comment. It has been added in abstract.
Line 35: please sort alphabetically in Key-words.
Thanks for your comment, please see line 33
Introduction:
Please reformat your entire manuscript to adhere to the MDPI journal's guidelines. For example, Line 39: ‘older adults.1 Anorexia of aging2’ to ‘older adults [1]. Anorexia of aging [2]’.
Thanks for your suggestion. The reference format has been corrected.
Please, you should more explain the literature reviews and backgrounds between dehydration and malnutrition factors. For example, you should add immune, inflammation, or nutrition point of view. I recommend that you should be added 4-5 paragraphs for backgrounds.
Thanks for your suggestion. We have added the consequences of malnutrition and dehydration in introduction, pleas see page 2 lines 61-67.
Method: Well-written
In Figure 1, please change the beginning and first letter of all words to uppercase.
Thank you for your feedback regarding the formatting in Figure 1. However, based on suggestions from another reviewer, we have removed Figure 1 from the manuscript and incorporated the relevant information directly into the text of the methods section for clarity and coherence. Please refer to the methods section for detailed descriptions.
Line 127: please delete whole sentence of line 127.
‘Nutritional assessment is described in a later section in greater detail.’
Thanks for your comment. It has been done.
Unfortunately, the line numbering in the version I received from the Nutrients Journal appears to differ from the one you referenced.
Results: Well-written
Line 183: Please delete sentence of ‘The CONSORT diagram is depicted in Figure 1.’. You already describe in Methods section.
Thanks for your suggestion. It has been deleted.
Line 172: DiA-S, Line 191: DIA-S. Please unify the terminology in whole manuscript.
Thanks for your suggestion. It has been in whole manuscript.
In Table 1: Please, the full name and abbreviation of SD in abbreviations section.
Thanks, it has been added. Please see line 215.
Line 194 and 212: Change ‘DiA-S, Depression in old Age’ to ‘DiA-S, depression in old age scale’, Please unify the terminology, too
Thanks for your suggestion. It has been corrected and the terminology has been unified.
You may revise all results to two decimal places in mean, standard deviation, etc., and three decimal places in statistical values (t, F value, p-value) are generally spelled out in academic writing. Please change in whole results and all Tables.
Thanks for your suggestion. It has been edited throughout the result section and tables.
Discussion
- You should add more applications in this study in Discussion section.
Thanks for your suggestion. We have added it (page 10, lines 378-388)
Round 2
Reviewer 2 Report
Comments and Suggestions for Authors
The authors corrected according to the instructions. Only in the abstract, "p" is missing from statistical significance. Plesae write p<0.001and p=0.903.
Author Response
Comments and Suggestions for Authors
The authors corrected according to the instructions. Only in the abstract, "p" is missing from statistical significance. Plesae write p<0.001and p=0.903.
Thank you for your comment. It has been added.